# Successful Pregnancy Outcome in a Patient Treated with Pembrolizumab and Exposed to Fluoro-Deoxyglucose (^18^F-FDG) PET/CT: Case Report and Review of Literature

**DOI:** 10.3390/biomedicines13010140

**Published:** 2025-01-09

**Authors:** Anna Lucia Mastricci, Felice Sorrentino, Elisa Giansiracusa, Erika Zanzarelli, Graziana Silvana De Lucia, Vincenza Fernanda Fesce, Luigi Nappi, Lorenzo Vasciaveo

**Affiliations:** 1Department of Obstetrics and Gynecology, Center of Maternal Fetal Medicine, Universitary Hospital, University of Foggia, 71122 Foggia, Italy; lucianamastricci@gmail.com (A.L.M.); elisag98@live.it (E.G.); erikazanzarelli5@gmail.com (E.Z.); grasil86@yahoo.it (G.S.D.L.); 2Department of Medical and Surgical Sciences, Institute of Obstetrics and Gynecology, University of Foggia, 71121 Foggia, Italy; felice.sorrentino.1983@gmail.com (F.S.); luigi.nappi@unifg.it (L.N.); 3Complex Operating Unit of Haematology, Universitary Hospital, University of Foggia, 71122 Foggia, Italy; vfesce@ospedaliriunitifoggia.it

**Keywords:** Hodgkin lymphoma, pregnancy, pembrolizumab, ^18^F-FDG PET/CT

## Abstract

**Background:** Hodgkin lymphoma (HL) is a common malignancy among women of reproductive age. Some pregnancies occur during oncological treatments or diagnostic follow-ups, often involving contraindicated procedures. HL is fluorodeoxyglucose-avid; therefore, its staging is generally performed with ^18^F-FDG PET/CT, a diagnostic method contraindicated during pregnancy. Immune checkpoint inhibitors (ICIs), such as pembrolizumab, are innovative therapies for relapsed HL (rHL) with significant efficacy. However, ICIs can impair immune tolerance, potentially increasing immune-related adverse events. The PD-1/PD-L1 pathway, targeted by pembrolizumab, plays a critical role in maternal–fetal immune adaptation, raising concerns about its safety during pregnancy. **Case Report:** We report the case of a 36-year-old woman diagnosed with rHL who unknowingly became pregnant during treatment with pembrolizumab and ^18^F-FDG PET/CT scans. The pregnancy was diagnosed at 24 weeks, after five cycles of pembrolizumab during the first two trimesters and an ^18^F-FDG PET/CT scan in the first trimester. Following multidisciplinary counseling, the pregnancy was closely monitored, culminating in the delivery of a healthy male infant at 37.5 weeks. **Conclusions**: This case highlights a favorable maternal–fetal outcome despite exposure to pembrolizumab and ^18^F-FDG PET/CT during pregnancy. Given the limited data on such exposures, case reports like this are essential for improving counseling and management strategies. Further research and registries are crucial to provide robust evidence for clinical decision-making in these complex scenarios.

## 1. Introduction

The incidence of pregnancies diagnosed in women with an oncological condition, or a history of oncological disease is approximately 1:1000. This incidence is expected to increase due to the rising average age of pregnant women, as advancing maternal age correlates with a higher risk of pre-pregnancy maternal conditions, including oncological diseases [1]. However, these pregnancies are often unintended and occur during therapeutic treatments. Therefore, it is highly advisable that young women with oncological conditions always receive counseling on contraception options [2]. Lymphoma is commonly diagnosed in adolescents and young adults and cases diagnosed during pregnancy are 1:3000 for Hodgkin lymphoma (HL) and 1:5000 for non-Hodgkin lymphoma (NHL) [3]. New treatments have improved the survival rate of patients with Lymphoma, leading to 90% after 5 years for early-stage Hodgkin’s Lymphoma [4,5]. HL diagnosed in pregnancy needs careful staging in order to carry out personalized treatment and minimize maternal and fetal complications.

The staging of this condition in non-pregnant patients is performed using ^18^F-fluorodeoxyglucose PET/CT (^18^F-FDG). Positron Emission Tomography/Computed Tomography with 18-fluoro-2-deoxy-2-fluoro-D-glucose (^18^F-FDG PET/CT) is a nuclear medicine technique that combines functional and anatomical information, providing valuable clinical insights. ^18^F-FDG PET/CT has been widely used in the staging of lymphoma, cutaneous melanoma, colorectal cancer, and other gastrointestinal malignancies. These findings can have a profound impact on patient management and appropriate follow-up. ^18^F-FDG PET/CT integrates the metabolic information provided by PET with the anatomical details from CT and offers several advantages: reduced image acquisition time, greater tumor localization accuracy, and whole-body staging in a single examination [6]. In pregnant patients affected by HL, we have to consider the effects of ionizing radiation and radiation dose on the embryo or fetus, according to gestational age. The guidelines of the American Society of Clinical Oncology (ASCO) established that ultrasound and magnetic resonance imaging are the imaging techniques to be preferred in pregnancy, safer but less sensitive. Small studies on ^18^F-FDG exposure in pregnancy, obtained from small and heterogeneous cohorts, have demonstrated the feasibility of ^18^F-FDG PET/CT(F-FDG) in pregnancy without fetal effects when fetal exposure dose is less than 50 mGy [7,8]. New treatment options are also available for patients with relapsed Hodgkin’s lymphoma (rHL). Pembrolizumab is effective in 70% of patients with rHL [9] by interfering with the interaction between PD-1 (programmed cell death protein 1) and its ligand (PD-L1) on T cells. PD-L1, expressed in tumor cells, binds to the PD-1 receptor on T cells, suppressing their function and promoting tumor growth. Pembrolizumab, by inhibiting the PD-1 protein, has shown excellent results in rHL [10]. However, the PD-1/PD-L1 balance is also essential for fetal development and pregnancy progression [11].

Currently, we have limited data on pregnancy outcomes in patients who have taken PD-1 inhibitors. One of the first published studies discusses the outcome of a pregnancy in a patient with metastatic melanoma treated with Nivolumab, which resulted in early fetal growth restriction (Early FGR) [12]. Indeed, there is little literature on pregnancy outcomes following the administration of PD-1 inhibitors, particularly in patients with Hodgkin’s lymphoma (HL) undergoing treatment with pembrolizumab. This makes it challenging to provide accurate counseling for these patients. In this article, we discuss the case of a patient with an unrecognized pregnancy until 24 weeks, who was diagnosed with relapsed Hodgkin’s lymphoma (rHL) and underwent diagnostic testing with ^18^F-FDG and pharmacological treatment with cycles of pembrolizumab.

## 2. Case Report

A multiparous patient, diagnosed with Hodgkin’s lymphoma (HL), nodular sclerosis subtype, in October 2020, was already undergoing treatment for type 1 diabetes with Insulin Lispro and Glargine. She underwent chemotherapy treatment according to the ABVD protocol (Doxorubicin, Bleomycin, Vinblastine, and Dacarbazine) from November 2020 to April 2021, with biweekly infusion cycles. Due to persistent disease after this regimen, second-line therapy with Dexamethasone, Cisplatin, and Cytarabine was initiated. The patient refused to undergo an autologous transplant. Subsequently, given disease progression, she underwent chemotherapy with Brentuximab and Bendamustine, followed by Brentuximab Vedotin (four cycles between January 2021 and June 2022). However, due to limited clinical response, treatment with pembrolizumab was started (11 cycles between September 2022 and May 2023). Following a positive pregnancy test performed due to prolonged amenorrhea, the patient was referred to our maternal–fetal medicine center. The last menstrual period was unknown, and an ultrasound dating the pregnancy performed on 6 June 2023, revealed a gestational age of approximately 24 weeks, estimating the last menstrual period (LMP) to be 20 December 2022. After determining the start of the gestational period, we reconstructed the treatments and diagnostic tests performed during the pregnancy (Figure 1):-Treatment with pembrolizumab from July 2022 to December 2022 (six cycles of 200 mg) with a possible sixth cycle occurring in the periconceptional period.-Treatment with pembrolizumab, 200 mg in February, March, April, May during the first and second trimester.-^18^F-FDG PET/CT on 18 January 2023, first trimester.

After a detailed evaluation of the fetal anatomy according to the SIEOG guidelines for the anomaly scan [13], excluding major malformations, a multidisciplinary counseling was immediately offered to the patient involving hematologists and neonatologists to detect the best maternal therapeutic and diagnostic strategy to ensure the best fetal–neonatal outcomes. The exclusion of major malformations, adequate fetal growth, and a functional fetal cardiac and maternal–fetal velocimetric evaluation, at the time of the examination (24 weeks), did not allow us to exclude the possible appearance of developmental malformations and/or appearance of placental insufficiency and fetal growth restriction later during the pregnancy; few studies on the effects of pembrolizumab in pregnancy allow us to exclude postnatal neurocognitive pathologies.

Expressing the desire to continue the pregnancy, the patient underwent ultrasound scans regularly to monitor fetal growth and to exclude any structural malformative pathologies. In addition, by scanning the lymph node stations, the evaluation of maternal clinical conditions was monitored. The timing and mode of delivery were discussed with hematologists, assessing the absence of the need to carry out the delivery before the 37th week of gestation, except in the case of maternal and/or fetal complications. In September 2023, at 37.5 weeks of gestation, the patient was hospitalized with a diagnosis of premature rupture of membranes (PROM). On the same day, she delivered a male newborn weighing 2650 g, with an APGAR score of 8 and 9 at 1 and 5 min, respectively.

The patient and newborn were discharged on the third day of the puerperium in good health. The newborn underwent a routine follow-up in accordance with the SIN guidelines for the first six months (Italian Society of Neonatology) [14]. No cognitive or neurobehavioral deficits were found during neonatal checks.

## 3. Discussion

In the literature, few cases are described of pregnant patients with Hodgkin’s lymphoma; pregnancy is not associated with an increased risk of Hodgkin’s lymphoma recurrence [1,15,16]. Managing pregnancy in oncology patients is challenging because, during pregnancy, especially in the first trimester, exposure to radiation used for cancer identification and staging must be minimized. Additionally, embryos and fetuses are highly vulnerable to teratogenic drugs, particularly during organogenesis (from the second to the eighth week of gestation) and later in pregnancy, which can affect the eyes, genitals, hematopoietic system, and central nervous system [17,18,19]. In our clinical case, the pregnancy was not recognized during the first trimester, and the patient was exposed to ^18^F-FDG PET/CT during routine rHL follow-ups. In the literature, few studies on exposure to ionizing radiation diagnostic techniques during pregnancy have shown that fetal exposures to doses below 100 mGy should not lead to complications such as malformations or intellectual disability (Table 1). In a study by Burton et al., involving nine patients at various pregnancy stages, it was confirmed that ^18^F-FDG can cross the placenta, accumulating in the gravid uterus and fetus. Despite the high risk of radiation damage susceptibility, all nine pregnancies resulted in the birth of healthy newborns [7], with self-induced fetal doses ranging from 1.2 to 8.2 mGy (Table 2).

Another study by Al Mansour et al., involving 18 pregnancies, evaluated fetal self-dose exposure levels equal to or higher than those in Burton’s study. In this case, the fetal self-dose ranged as follows: in the second trimester, 5.7–15.8 mGy (CT-Expo method) and 5.1–11.6 mGy (VirtualDose); in the third trimester, 7.9–16.6 mGy (CT-Expo method) and 6.1–10.7 mGy (VirtualDose) (Table 3). Again, all pregnancies resulted in full-term births without malformations [8].

In our case, the pregnancy was exposed to an estimated fetal dose of 11 mGy, consistent with the data from Burton and Al Mansour’s studies. Pembrolizumab was used during the first six months of pregnancy. Pembrolizumab belongs to the immune checkpoint inhibitor (ICI) family, a class of biological drugs widely used to treat various malignancies [20]. These drugs, such as monoclonal antibodies that block CTLA-4 (cytotoxic T-lymphocyte-associated protein 4), LAG-3 (lymphocyte activation gene 3), and PD-1 or its ligand (PD-L1), restore T-cell immune responses in different cancer types [21]. However, ICIs can also interfere with immune tolerance, increasing the risk of immune-related adverse events, including autoimmunity [22]. At the syncytiotrophoblast and extravillous cytotrophoblast level, there is a high expression of PD-L1, which plays a crucial role in immune tolerance by binding to PD-1 receptors on maternal cells. The maternal–fetal immune tolerance process during pregnancy involves complex interactions of immunomodulatory mechanisms, where the maternal immune system adapts to the presence of the semi-allogeneic fetus, preventing immune rejection and maintaining the balance essential for a successful pregnancy [23]. Notably, the expression of immune checkpoints, particularly PD-1 on T cells at the maternal–fetal interface, increases as pregnancy progresses to prevent fetal immune rejection [24]. Anticancer immunotherapies could theoretically disrupt this maternal–fetal immune balance, posing risks of pregnancy complications [11,25]. In studies on cynomolgus monkeys, injecting high doses of the anti-PD-1 agent Nivolumab (>10 times the clinical dose) resulted in an increased risk of fetal growth restriction, preterm birth, and fetal and neonatal death compared to placebo [26]. Pembrolizumab is a humanized monoclonal antibody targeting the programmed cell death protein-1 (PD-1) receptor expressed on T lymphocytes’ cell surfaces and is used to treat various malignancies. Being an IgG4 antibody, it can cross the placenta and reach the fetus. Blocking the PD-L1 signaling pathway may impair fetal tolerance, increasing the likelihood of miscarriage or intrauterine fetal death [27,28]. To date, no studies have been conducted on the use of pembrolizumab during pregnancy, but interference with the PD-1 system could increase the rate of complications. A few cases in the literature report on pembrolizumab use in relation to pregnancy (Table 4):-A 25-year-old patient with classical Hodgkin’s lymphoma. Initially treated with chemotherapy and autologous stem cell transplantation, followed by 21 cycles of pembrolizumab. Two years after treatment, an unexpected pregnancy occurred, resulting in a healthy live birth without maternal or fetal complications [29].-A 40-year-old nulliparous patient with advanced-stage melanoma diagnosed in the first trimester. Despite maternal and fetal risks, the patient decided to continue the pregnancy and was treated with pembrolizumab from 21 to 27 weeks of gestation. A cesarean section was performed at 28 weeks, delivering a newborn without complications. Maternal outcomes, however, showed disease progression despite treatment [30].-A 35-year-old patient with gastric cancer, treated with neoadjuvant chemotherapy and subsequent surgery. During follow-up, metastases were detected, and the patient underwent radiotherapy followed by immunotherapy. During these treatments, a pregnancy was diagnosed, and the patient decided to continue it. Fetal growth restriction and dilated fetal bowel led to cesarean delivery at 35 weeks. Necrotizing enterocolitis occurred, requiring surgical resection, while the patient continued pembrolizumab treatment and achieved positive responses [31].

Recent studies [32,33] have considered a larger series of ICIs used during pregnancy, suggesting that the successful pregnancy outcomes despite the pembrolizumab treatment might be related to varying IgG concentrations to which the fetus is exposed during pregnancy. If IgG levels are low in the first and second trimesters, organogenesis may not be affected.

The main strengths of our article include the rarity of the association between pembrolizumab treatment and ^18^F-FDG PET/CT imaging results, as well as the detailed presentation of a unique case that enriches the discussion on maternal–fetal outcomes. This case contributes to the body of literature by providing valuable insights that could support future research. However, the study has significant limitations, including the lack of complete long-term pediatric follow-up data and an adequate discussion of ethical and psychological aspects. We publish this case report to improve medical clinical management and counseling regarding the safety of diagnostic tests and oncological therapies in patients of reproductive age with oncological conditions. Certainly, further data are needed, and it is desirable to create registries for these drugs where data can be reported, correlating gestational ages and treatments received with maternal, fetal, neonatal, and pediatric outcomes and follow-ups.

## 4. Conclusions

Currently, very few cases of pregnancies occurring during or immediately after the administration of pembrolizumab are reported in the literature. The management of pregnancy complicated by oncological diseases, like Hodgkin’s lymphoma, represents a great challenge for the obstetrician in clinical management difficulties and professional ethics. A multidisciplinary management of these pregnancies, involving specialists in maternal–fetal medicine, oncologists, hematologists, anesthetists, neonatologists, psychologists, and detailed counseling is essential for patients to decide if to interrupt oncological treatments or terminate the pregnancy.

In our case, multidisciplinary counseling was ethically complex given the advanced stage of maternal pathology and fetal good condition in advanced gestational age at the diagnosis of pregnancy. The maternal choice to continue the pregnancy was decisive for her own evolution and pregnancy outcome.

## Figures and Tables

**Figure 1 biomedicines-13-00140-f001:**
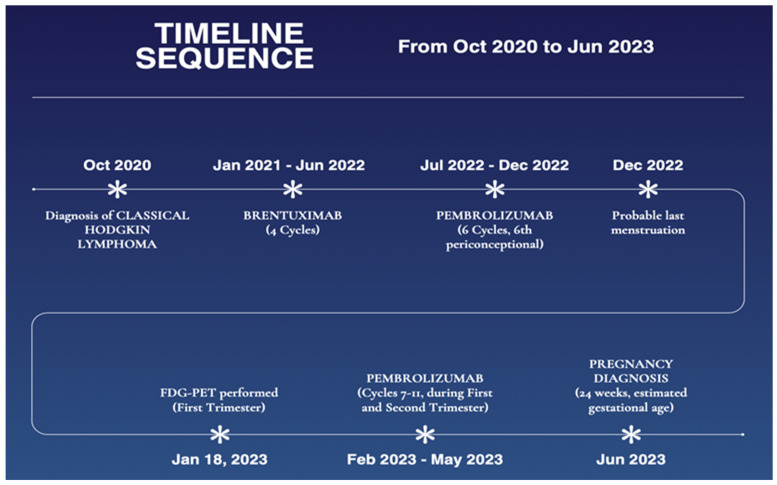
Timeline sequence. This timeline illustrates the sequence of events from October 2020 to June 2023, documenting the diagnosis, treatment milestones, and key personal developments of our patient. It highlights pivotal moments such as the diagnosis of Classical Hodgkin lymphoma, pembrolizumab treatments, FDG-PET examination during pregnancy, and the pregnancy diagnosis at 24 weeks of estimated gestational age.

**Table 1 biomedicines-13-00140-t001:** Studies on Exposure During Pregnancy (* Fetal-self dose estimated using CT-Expo 2.7 during II and III trimesters; ° Fetal-self dose estimated using Virtual Dose during II and III trimesters).

	Burton et al. [7]	Al Mansour et al. [8]	Our Case
N° of pregnancy	9	18	1
Fetal-self dose	1.2 to 8.2 mGy	5.7–15.8 to 7.9–16.6 mGy (*) 5.1–11.6 to 6.1–10.7 mGy (°)	11 mGy

**Table 2 biomedicines-13-00140-t002:** The study of Burtonet Al. provides data on fetal radiation dose estimates during imaging procedures involving ^18^F-FDG PET. It focuses on the fetal self-dose and total dose, as well as the contribution of the size-specific dose estimate (SSDE) to the total dose. All the patients were expecting a single fetus and the gestational age ranged from 12 to 16 weeks.

Patients	Gestational Age (Week)	SSDE Range (mGy)	^18^F-FDG Fetal Self-Dose Range (mGy)	^18^F-FDG Fetal Total Dose Range (mGy)	SSDE + ^18^F-FDG Fetal Self-Dose Range (mGy)
9	From 12 to 36	1.0–6.9	0.0017–2.18	0.0034–2.35	1.2–8.2

**Table 3 biomedicines-13-00140-t003:** The study of Al Mansour et al. described data on the estimated fetal and uterine radiation exposure during CT scans for different medical indications in pregnant patients, using CT-Expo 2.7 and VirtualDose version is 2.8. Different gestational ages, ranging from 16 weeks to 33 weeks, are included and indications vary widely. Only one of all patients was expecting triplets but she was considered as expecting a single fetus for the dosimetry analysis. (* During II Trimester; ° During III Trimester).

Patients	Gestational Age (Weeks)	Average Patient Weight (kg)	Estimated Fetal Self-Dose CT-Expo Method (mGy) *	Estimated Fetal Self-Dose Virtual Dose (mGy) *	Estimated Fetal Self-Dose CT-Expo Method (mGy) °	Estimated Fetal Self-Dose Virtual Dose (mGy) °	Adverse Fetal Outcomes
18	From 16 to 33	71.2	5.7–15.8	5.1–11.6	7.9–16.6	6.1–10.7	/

**Table 4 biomedicines-13-00140-t004:** Four cases of pregnant women treated with pembrolizumab for different tumors. Indications, dose of treatment (* every 3 weeks), characteristics of delivery, and fetal and maternal outcomes are described for each case. (GA, gestational age; VD, Vaginal Delivery; CS, Cesarean Section; R, Remission; PD, Progressive Disease; PR, Positive Response; NEC, Necrotizing Enterocolitis; ? Unknown).

	Indication	Cycles	Dose	Pregnancy Timing	Delivery	GA at Delivery	Maternal Outcome	Adverse Fetal Outcome	Neonatal Weight (g)	Percentile Weight	Apgar Scores
1	Classical Hodgkin Lymphoma	21	?	2 years after treatment	VD	?	R	/	3100	?	8–9
2	Advanced Malignant Melanoma	3	200 mg *	21–27 weeks of gestation	CS	28 weeks	PD	/	1291	>90°	7–8
3	Gastric Carcinoma	?	200 mg *	Treatment during pregnancy 2 years after treatment	CS	35 weeks	PR	NEC	1685	<10°	9–10
4	Classical Hodgkin Lymphoma	10	200 mg *	During pregnancy	VD	37 weeks 5 days	PD	/	2650	<10°	8–9

## Data Availability

The original contributions presented in the study are included in the article. Further inquiries can be directed to the corresponding author.

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
