# Peer review of "Successful Pregnancy Outcome in a Patient Treated with Pembrolizumab and Exposed to Fluoro-Deoxyglucose (18F-FDG) PET/CT: Case Report and Review of Literature"

_biomedicines, 2025, doi:10.3390/biomedicines13010140_

Round 1
Reviewer 1 Report
Comments and Suggestions for Authors
Comments:
The authors reported a case that a 36-year-old woman diagnosed with rHL who unknowingly became pregnant during treatment with pembrolizumab and 18F-FDG PET/CT scans. Following multidisciplinary counseling, the pregnancy was closely monitored, culminating in the delivery of a healthy male infant at 37.5 weeks. The authors also reviewed current studies for future use.
Major:
1. What information is available regarding the 18F-FDG PET/CT in this case, like the exposure dose, 18F-FDG PET/CT image? More information about the disease condition of rHL may be beneficial.
2. The information in the case report (lines 70-85) needs to be better organized. Providing more organized medication information would be beneficial.
Minor:
- For the phrase “18F-FDG PET/CT,” ensure consistency throughout the manuscript. Some instances are written as “F-FDG PET/CT.”
2. Line 55: The abbreviation "rLH" is incorrect; it should be "rHL." Please correct this and any other instances of the same typo throughout the manuscript.
3. Line 55: The word " Prem-bolizumab " is incorrect; it should be " Pem-bolizumab." Please correct this and any other instances of the same typo throughout the manuscript.
4. Line 56, PD-1: Stands for "Programmed Cell Death Protein 1," not "cell death protein1." PD-L1: Stands for "Programmed Death-Ligand 1," Please correct this and any other instances of the same typo throughout the manuscript. Lines 156 and 173: There is a repeated definition of the abbreviation. Please remove the redundancy.
5. Line 61, The word "evolution" is inappropriate. Maybe use “progression”
6. Line 105, is it better to be more formal? like “weighed 2650 g, male, and had APGARs of 8 and 9 at 1 min and 5 min, respectively”.
7. Lines 178-179: The statement 'To date, no studies have yet been conducted' may not be accurate. For example, this paper (PMID: 34487972) discusses some clinical studies.
Reviewer 2 Report
Comments and Suggestions for Authors
This article (Manuscript ID: biomedicines-3405040) presented the case of a 36-year-old woman diagnosed with rHL who unknowingly became pregnant during treatment with pembrolizumab and 18F-FDG PET/CT scans. The pregnancy was closely monitored, culminating in the delivery of a healthy male infant at 37.5 weeks. It claimed that this case highlights a favorable maternal-fetal outcome despite exposure to pembrolizumab and 18F-FDG PET/CT during pregnancy.
The research topic is within the scope of this journal. The novelty of this report is somewhat good. However, the writing, data assessment, and organization of this text need careful improvement. This paper should be made careful revisions.
Best wishes and kind regards.
Comments to the authors
Major issues:
Q1. In the abstract section, what’s F-FDG PET/CT? Line 22, e report?
Q2. The introduction should be improved carefully. The organization of the introduction should be reconsidered. Lines 36 to 39 and lines 62 to 63, there are just two short paragraphs. Also, the ligand is presented as PDL1. Maybe, it could be PD-L1. Moreover, PD1 should be PD-1.
Q3. More citations should be added to this report.
Q4. What’s the criterion for the selection of multiparous patient?
Q5. Figure 1 should be improved carefully. The presentation of the tables should be improved.
Q6. The discussion section should be checked and improved.
Q7. What is the source of pembrolizumab?
Q8. Please check the writing and expression of the references.
Q9. The organization of the results and discussion section has several shortcomings that need to be addressed throughout the submission process. There are issues with the text's writing. Problems are present in sentence structure, verb tense, and the construction of subordinate clauses. The writing should be improved, and it is recommended to carefully revise the expressions.
Q10. The quality of English needs to be improved. I admit that the author is not a native English speaker. However, the manuscript was poorly written and the quality of the language compromised the reader's understanding many times. I recommend working with a language professional or a language polishing company to improve the manuscript.
Comments on the Quality of English Language
The quality of English needs improving. I acknowledge that the authors are non-native English speakers. However, the manuscript is not well written, and the language quality impairs the reader’s understanding on several occasions. I recommend improving the manuscript with a language professional or a language polishing company.
Round 2
Reviewer 1 Report
Comments and Suggestions for Authors
N/A